# Trends in opioid prescribing practices in South Korea, 2009–2019: Are we safe from an opioid epidemic?

Noo Ree Cho[1], Young Jin Chang[1], Dongchul Lee[1], Ji Ro Kim[1], Dai Sik Ko[2]*, Jung Ju Choi[1]*

**1** Department of Anesthesiology and Pain Medicine, Gachon University Gil Medical Center, Incheon, Republic of Korea, **2** Division of Vascular Surgery, Department of Surgery, Gachon University Gil Medical Center, Incheon, Republic of Korea

* daisik.ko@gilhospital.com (DSK); jjchoi2@gilhospital.com (JJC)

**Data Availability Statement:** All relevant data are within the manuscript and its Supporting Information files.

## Abstract

Opioid prescribing data can guide regulation policy by informing trends and types of opioids prescribed and geographic variations. In South Korea, the nationwide data on prescribing opioids remain unclear. We aimed to evaluate an 11-year trend of opioid prescription in South Korea, both nationally and by administrative districts. A population-based cross-sectional analysis of opioid prescriptions dispensed nationwide in outpatient departments between January 1, 2009, and December 31, 2019, was conducted for this study. Data were obtained from the Health Insurance Review & Assessment Service. The types of opioids prescribed were categorized into total, strong, and extended-release and long-acting formulation. Trends in the prescription rate per 1000 persons were examined over time nationally and across administrative districts. There are significant increasing trends for total, strong, and extended-release and long-acting opioid prescriptions (rate per 1000 persons in 2009 and 2019: total opioids, 347.5 and 531.3; strong opioids, 0.6 and 15.2; extended-release and long-acting opioids, 6.8 and 82.0). The pattern of dispensing opioids increased from 2009 to 2013 and slowed down from 2013 to 2019. The rate of opioid prescriptions issued between administrative districts nearly doubled for all types of opioids. Prescription opioid dispensing increased substantially over the study period. The increase in the prescription of total opioids was largely attributed to an increase in the prescription of weak opioids. However, the increase in prescriptions of extended-release and long-acting opioids could be a future concern. These data may inform government organizations to create regulations and interventions for prescribing opioids.

## Introduction

The United States (US) is battling an opioid overdose epidemic. A total of 70,237 drug overdose deaths occurred in 2017: an age-adjusted rate of 21.7 per 100,000 persons [1]. Prescription and/or illicit opioids were involved in approximately two thirds (47,600) of these deaths.

**Funding:** DSK; National Research Foundation of Korea(NRF) grant (NRF-2020R1A2C1102433), Young Medical Scientist Research Grant through the Daewoong Foundation (DY20111P) NRC; Korea Medical Institute. The funders had no role in study design, data collection and analysis, decision to publish, or preparation of the manuscript.

**Competing interests:** The authors have declared that no competing interests exist.

Among opioid-involved deaths, the category of synthetic opioids, which includes illicitly manufactured fentanyl, was the most common cause (28,466 deaths). Prescription opioids, which include natural and semi-synthetic opioids (e.g., oxycodone and hydrocodone) and methadone, were the second most common cause, with 17,029 deaths. Premature deaths from opioid overdose partly decreased recent US life expectancy [2]. On August 10, 2017, the US government declared the opioid crisis a national public health emergency to curb a rapidly escalating public health crisis [3].

Deaths from prescription opioid-related overdose have increased in parallel with increases in opioids prescribed in the US, which is a 4-fold increase from 1999 to 2010 [4]. Changes in US governmental policies in the late 1990s may have contributed to this increase. Pain management ideology underwent a change in treating pain as a fifth vital sign [5] and government regulations on the prescription of opioids relaxed. The increase in the prescription of opioids was mostly due to an increase in the use of opioids to treat chronic noncancer pain [6, 7]. As a result, the number of opioids prescribed per person peaked in 2010 and subsequently decreased thereafter [8].

The misuse of prescription opioids and related mortality have not been an issue in Korea [9]. However, most of the aforementioned synthetic opioids are also available in South Korea through outpatient departments of primary, secondary, and tertiary hospitals. A recent study reported that patients with chronic use (over 90 days of continuous supply) of weak and strong opioids increased between 2002 and 2015 [10]. This suggests that the prevalence of chronic opioid use is increasing, which might lead to opioid misuse. However, there are no reports showing the opioid outpatient prescription trends, which are associated with opioid misuse. Accordingly, we aimed to examine opioid outpatient prescription trends from 2009 through 2019 using the following strategies: (1) classification of opioids by potency and formula; (2) changes in the opioid outpatient prescriptions nationally each year; (3) geographical differences in the opioid outpatient prescriptions.

## Materials and methods

### Data source

We obtained data from outpatient prescription records provided by the Health Insurance Review and Assessment Service from January 1, 2009, to December 31, 2019. This database provides the number of opioid prescriptions dispensed from the outpatient department of primary, secondary, and tertiary hospitals covering the entire population in South Korea. We excluded cough and cold formulations containing opioids and inpatient opioid prescriptions (Fig 1). Since the records did not contain any identifying information, this study was exempt from ethical review by the Gachon University Gil Medical Center's ethics review board.

### Definitions and variables

We described three key measures at the national and 17 administrative districts: weighted annual prescription rates per 1000 persons prescribed at outpatient departments for (1) total opioids, (2) strong opioids, and (3) extended-release and long-acting (ER/LA) opioids. The three annual prescribing rates were calculated as population-based rates by weighting raw values to national and administrative district populations for each study year [11].

We defined strong opioids as those which are equivalent to or higher than morphine's potency—the morphine milligram equivalents (MME) conversion factor is equal to or higher than 1 and is available in South Korea. The strong opioids were morphine, hydrocodone, fentanyl (including transdermal patches), hydromorphone, and oxycodone. The weak opioids were codeine, dihydrocodeine, tapentadol, and tramadol. Prescription formulations were

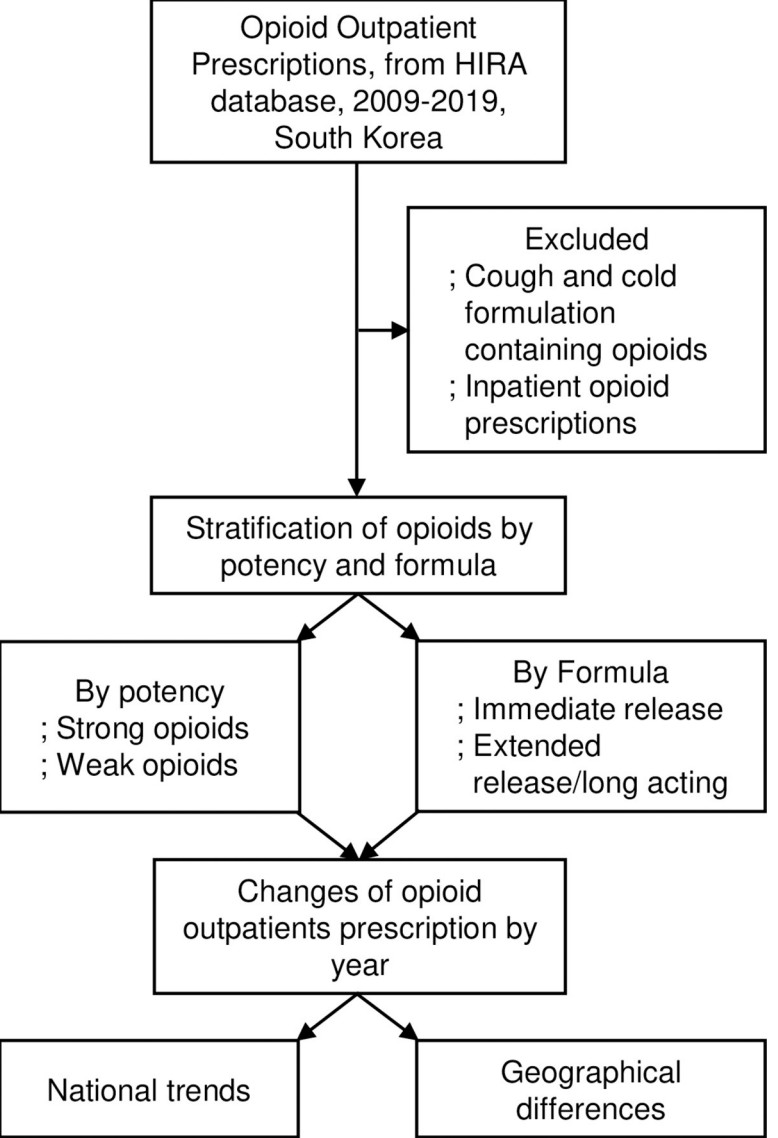

**Fig 1. Flow chart of study design.** Abbreviation: HIRA, Health Insurance Review and Assessment.

categorized as (1) immediate-release opioids or (2) ER/LA opioids, such as transdermal fentanyl citrate or as an ER/LA formulation of an immediate-release drug.

## Statistical analysis

Data were abstracted and descriptive analyses were completed using R (version 3.6.0, Vienna, Austria). We reported the temporal trends of national and administrative district averages for each variable using Joinpoint regression analysis (version 4.8.0.1; National Cancer Institute) [12]. Joinpoint uses log-linear regression to fit the simplest trend of the data and calculate percentage changes. Trends spanning from 2009 to 2019 were computed as the mean annual percentage change. Shorter time segment trends were computed as the annual percentage change. Annual percentage change and mean annual percentage change for each variable are expressed as the percentage change with a 95% Confidence Interval (CI). The terms *increase* and *decrease*

refer to an annual percentage change significantly different from zero. All hypothesis testing was two-tailed, with statistical significance set at two-sided $P < .05$.

Administrative district-level geographic inequality in opioid-prescribing attributes was quantified by comparing the 10th and 90th percentiles for each variable among all administrative districts [13]. Using the 90th and 10th percentiles instead of the maximum and minimum values reduced the ability of outliers to skew results for each variable. The difference between these two percentiles was used to indicate the degree of absolute geographic inequality, representing the absolute magnitude of the gap between high- and low-prescribing administrative districts for each variable. The ratio between the 90th and 10th percentiles was used to assess the relative degree of geographic inequality between administrative districts for each variable.

We used the R package, "Kormap," to download South Korea's map that shows all the administrative districts. The "Kormap" package is a transformation of the shape file of South Korea's map from the Statistical Geographic Information Service for utilization in R. The "Tmap" R package was used to visualize the administrative districts' disparity in opioid prescription in South Korea.

## Results

From 2009 to 2019, an average of 23.2 million opioid prescriptions were issued from outpatient departments in South Korea (Table 1). The total opioid prescriptions in 2019 was equivalent to 53% of the South Korean population.

### Annual amount of total opioid prescriptions per person

Single-year values for the number of opioid prescriptions increased continuously by year (Fig 2). The rate per 1000 persons of total opioid prescriptions was 347.5 in 2009 and 531.3 in 2019 (Table 1). The mean rate per 1000 persons of total opioid prescriptions by administrative

**Table 1. Annual opioid prescriptions for the three key measures in South Korea, 2009–2019[a].**

| Year | Total Opioid prescriptions, n | Total Opioid prescriptions, rate per 1000 persons | Strong Opioid prescriptions, n, (%)[b,c] | Strong Opioid prescriptions, rate per 1000 persons[c] | Prescription for ER/LA formulation, n, (%)[b] | Prescription for ER/LA formulation, rate per 1000 persons |
|---|---|---|---|---|---|---|
| 2009 | 17,135,625 | 347.5 | 31,721 (0.2%) | 0.6 | 335,304 (2.0%) | 6.8 |
| 2010 | 18,038,145 | 364.0 | 36,131 (0.2%) | 0.7 | 307,828 (1.7%) | 6.2 |
| 2011 | 19,688,339 | 394.3 | 50,376 (0.3%) | 1.0 | 675,982 (3.4%) | 13.5 |
| 2012 | 22,277,939 | 443.8 | 170,430 (0.8%) | 3.4 | 1,063,762 (4.8%) | 21.2 |
| 2013 | 23,854,168 | 473.0 | 362,947 (1.5%) | 7.2 | 2,164,858 (9.1%) | 42.9 |
| 2014 | 24,371,569 | 480.3 | 439,157 (1.8%) | 8.6 | 3,031,196 (12.4%) | 59.7 |
| 2015 | 24,591,434 | 482.0 | 552,740 (2.2%) | 10.8 | 3,284,740 (13.4%) | 64.4 |
| 2016 | 25,363,266 | 495.2 | 639,640 (2.5%) | 12.5 | 3,682,383 (14.5%) | 71.9 |
| 2017 | 25,668,017 | 499.7 | 695,222 (2.7%) | 13.5 | 3,912,110 (15.2%) | 76.2 |
| 2018 | 26,228,556 | 508.2 | 732,129 (2.8%) | 14.2 | 4,012,150 (15.3%) | 77.7 |
| 2019 | 27,474,634 | 531.3 | 785,534 (2.9%) | 15.2 | 4,242,634 (15.4%) | 82.0 |
| Mean (SD) | 23,153,790 (3,442,847) | 456.3 (61.3) | 408,730 (295,279) | 8.0 (5.7) | 2,428,450 (1,566,869) | 47.5 (30.3) |

Abbreviations: ER/LA, extended-release and long-acting

[a]Sejong-si was included after 2011 as it has been incorporated into the administrative district since 2012.

[b]Percentage of the total opioid prescriptions.

[c]A strong opioid prescription was defined as an opioid with a morphine milligram equivalent that is equal to or greater than that of morphine.

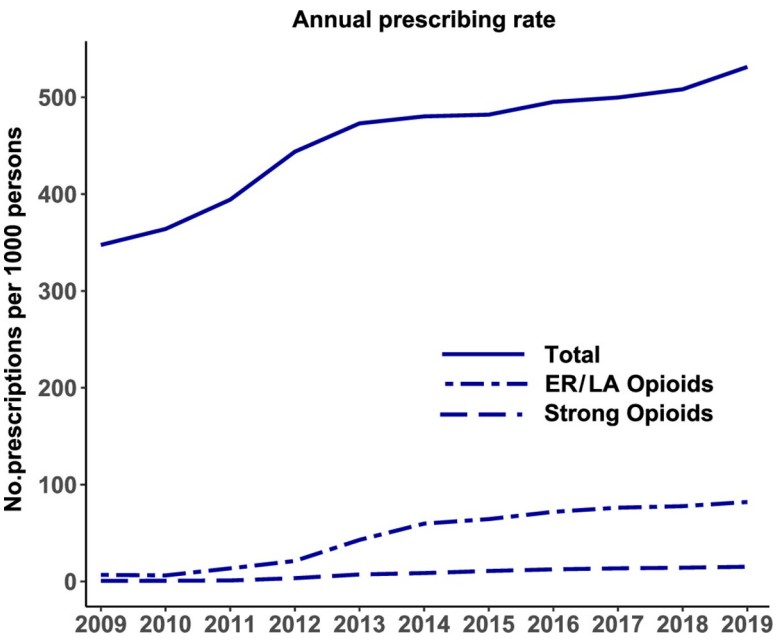

**Fig 2. Annual opioid prescribing rates per 1000 persons for the three key measures (total, strong, and ER/LA opioids).**

districts had a geographic inequality gap of 315.7 per 1000 persons between the 90th percentile and 10th percentile values in 2019 (Table 2). The ratio of the 90th and 10th percentile values was 1.7 in 2019, representing a 1.7-fold variation among administrative districts (Table 2). Over 11 years, the rate per 1000 persons of total opioid prescriptions increased by a mean (SD) of 51.0% (19.1%) among administrative districts, and the absolute geographic inequality increased from 303.5 to 315.7, yet relative geographic inequality decreased from 2.2 to 1.7 (Table 2). The administrative districts with a rate per 1000 persons of total opioids prescribed exceeding 750 were Jeollabuk-do and Jeollanam-do in 2019 (Fig 3A and S1 Table).

Joinpoint analysis indicated that the national rate per 1000 persons increased annually by 8.3% (95% CI, 6.2. -10.5%; $P < 0.001$) from 2009 to 2013 and 1.8% (95% CI, 0.8%-2.7%; $P < 0.001$) from 2013 to 2019 (S1 Table). By 2013, the rate per 1000 persons of total opioid prescriptions increased; this increase slowed down from 2013 to 2019. This pattern was observed in all administrative districts except Chungcheonam-do, Gyeongsangbuk-do, and Jeju-do (Fig 3A and S1 Table). These administrative districts showed a distinct pattern of increased and decreased opioid prescriptions with an annual percent change (APC) as follows: -1.1% (95% CI, -2.4%-0.1%, $P < 0.001$) in Chungcheonam-do from 2013 to 2019; -0.6% (95% CI, -1.1%—0.1%, $P < 0.001$) in Gyeongsangbuk-do from 2012 to 2019; and -0.2% (95% CI, -2.2%-1.8%, $P < 0.001$) in Jeju-do from 2014 to 2019. Among all administrative districts, Busan and Incheon showed the largest increase in the average APC of 6% (95% CI, 4.8%-7.1%, P < 0.001) and 6% (95% CI, 4.5%-7.6%), respectively.

## Strong opioid prescriptions

Over 11 years, a mean (SD) of 8.0 (5.7) strong opioid prescriptions per 1000 persons were issued from outpatient departments in South Korea (Table 1). The rate of strong opioid prescriptions per 1000 persons increased markedly from 0.6 in 2009 to 15.2 in 2019 (Fig 2 and

**Table 2. Summary of trends in opioids prescribed for administrative districts in South Korea, 2009–2019[a].**

| Characteristics by Year | Mean (SD) [Median][b] | Percentile | | Geographical Inequality | | Administrative Districts with statistically significant change | |
|---|---|---|---|---|---|---|---|
| | | 10th | 90th | Absolute[c] | Relative[d] | Decrease[e] | Increase[f] |
| **Total amount of opioids prescribed, rate per 1000 persons** | | | | | | | |
| 2009 | 401.4 (121.5) [379.5] | 254.2 | 557.7 | 303.5 | 2.2 | | |
| 2019 | 590.4 (138.7) [586.7] | 446.7 | 762.4 | 315.7 | 1.7 | | |
| Change (2009–2019), % | 51.0 (19.1) [50.2] | 30.0 | 75.7 | | | | |
| Trend (2009–2019), No. (%) | | | | | | | 16 (100.0) |
| **Strong opioid prescribed, rate per 1000 persons** | | | | | | | |
| 2009 | 0.6 (0.5) [0.7] | 0.1 | 1.0 | 0.9 | 10 | | |
| 2019 | 14.5 (5.3) [13.8] | 9.7 | 23.8 | 14.1 | 2.5 | | |
| Change (2009–2019), % | 9001.6 (19468.2) [2885.7] | 1164.6 | 16216.1 | | | | |
| Trend (2009–2019), No. (%) | | | | | | | 16 (100.0) |
| **Prescription for ER/LA formulation, rate per 1000 persons** | | | | | | | |
| 2009 | 7.9 (2.6) [7.7] | 5.1 | 10.4 | 5.3 | 2.0 | | |
| 2019 | 86.0 (16.9) [88.6] | 64.9 | 107.1 | 42.2 | 1.7 | | |
| Change (2009–2019), % | 1072.8 (348.7) [1151.3] | 667.4 | 1422.1 | | | | |
| Trend (2009–2019), No. (%) | | | | | | | 16 (100.0) |

Abbreviations: ER/LA, extended-release and long-acting

[a]Total 16 administrative districts were involved except Sejong-si, which has been incorporated into the administrative district since 2012.

[b]Mean was calculated from the values from 16 Administrative Districts. This mean does not reflect Korea's national value.

[c]Measure of absolute geographic inequality was calculated by subtracting the 10th percentile from the 90th percentile

[d]Measure of relative geographic inequality was calculated as the ratio of the 90th percentile to the 10th percentile.

[e]Indicates that a trend was significantly different from zero at the α = .05 level ($P < .05$) and that the mean annual percentage change had a negative value according to the Joinpoint regression analysis.

[f]Indicates that a trend was significantly different from zero at the α = .05 level ($P < .05$) and that the mean annual percentage change had a positive value according to the Joinpoint regression analysis.

Table 1). The mean rate per 1000 persons of strong opioid prescriptions by administrative districts had a geographic inequality gap of 14.1 between the 90th and the 10th percentile values in 2019 (Table 2). The ratio of the 90th and 10th percentile values was 2.5 in 2019 (Table 2). Over 11 years, the rate per 1000 persons of strong opioid prescriptions increased by a mean (SD) of 9001.6% (19468.2%) among administrative districts and absolute geographic inequality increased from 0.9 to 14.1, yet relative geographic inequality decreased from 10 to 2.5 (Table 2). Administrative districts with a rate per 1000 persons of strong opioid prescriptions exceeding 24 were Seoul and Jeollabuk-do in 2019 (Fig 3B and S2 Table).

Joinpoint analysis indicated that the rate per 1000 persons of national strong opioid prescriptions increased annually by 116.0% (95% CI, 60.3%-191.1%; $P < 0.001$) from 2009 to 2013 and by 12.8% (95% CI, 7.2%-18.8%; $P < 0.001$) from 2013 to 2019 (S2 Table). By 2013, the rate per 1000 persons of strong opioid prescriptions increased; the increase subsequently slowed from 2013 to 2019. This pattern was observed in all administrative districts, except Gyeongsangbuk-do (Fig 3B and S2 Table). This administrative district showed a decrease with the APC, namely -0.7% (95% CI, -6.0%-5.0%, $P < 0.001$) from 2014 to 2019. Among all administrative districts, Jeju-do showed the largest increase in the average APC, namely 129.1% (95% CI, 27.6%-311.2%, $P < 0.001$) (S2 Table).

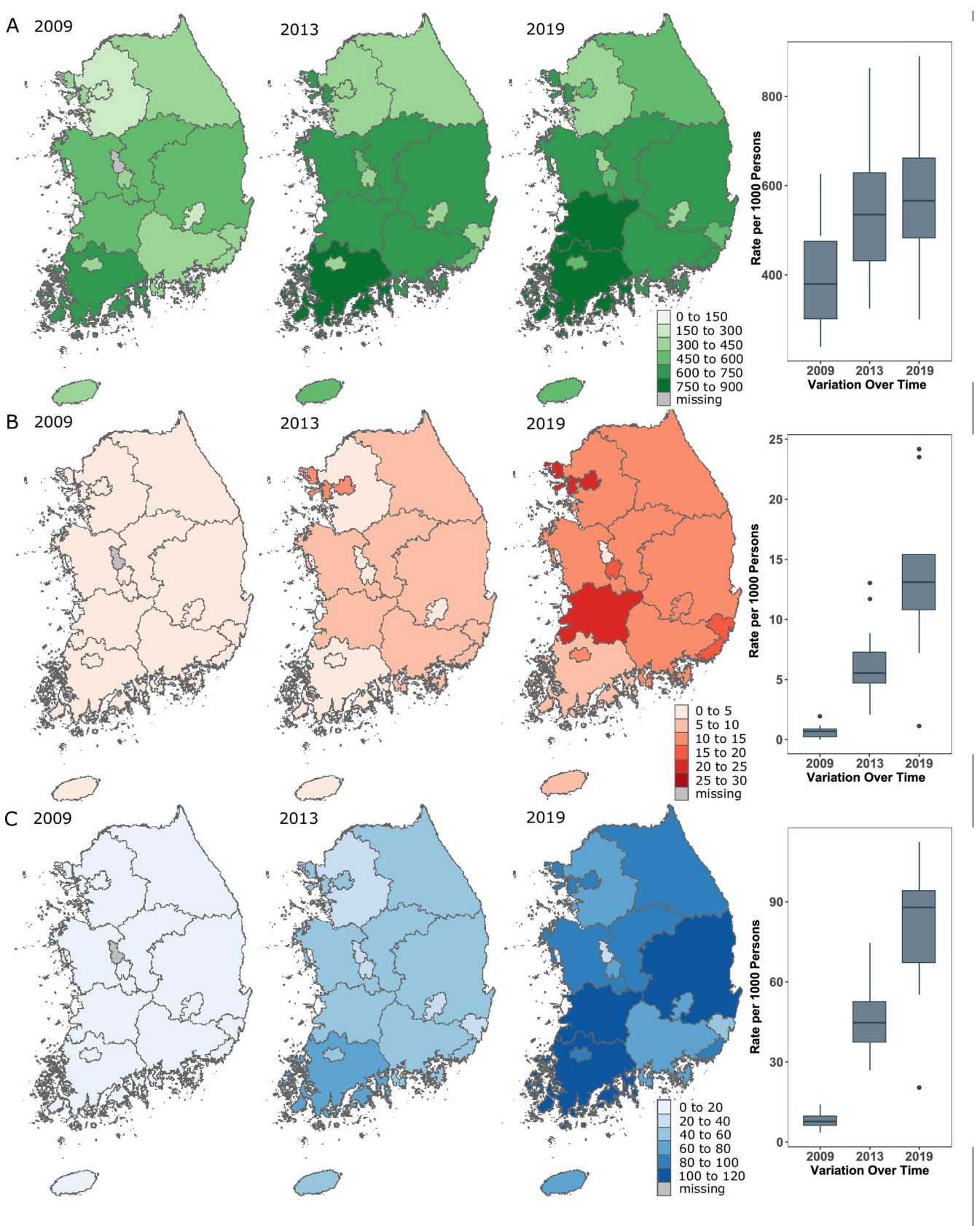

**Fig 3. Changes in annual rate of opioid prescriptions per 1000 persons from years 2009, 2013, and 2019.** (A) The rate of prescription of total opioids. (B) The rate of prescription of strong opioids. (C) The rate of prescription of ER/LA opioids. Data courtesy of OpenStreetMap (openstreetmap.org: OpenStreetMap contributors). Abbreviation: ER/LA, extended release/long acting.

### Prescriptions with ER/LA formulations

Between 2009 and 2019 (11 years), a mean (SD) of 47.5 (30.3) per 1000 persons of ER/LA opioid prescriptions were issued annually by outpatient departments in South Korea (Table 1). The rate of ER/LA opioid prescriptions per 1000 persons sharply increased from 6.8 in 2009 to 47.5 in 2019 (Fig 2 and Table 1). The mean rate per 1000 persons of ER/LA opioid prescriptions by administrative districts had a geographic inequality gap of 42.2 between the 90th percentile and 10th percentile values in 2019 (Table 2). The ratio of the 90th and 10th percentile values was 1.7 in 2019 (Table 2). Over 11 years, the rate per 1000 persons of ER/LA opioid prescriptions increased by a mean (SD) of 1072.8% (348.7%) among administrative districts and the absolute geographic inequality increased from 5.3 to 42.2, yet relative geographic inequality decreased from 2.0 to 1.7 (Table 2). Administrative districts with rates per 1000 persons of ER/LA opioid prescriptions exceeding 100 were Jeollabuk-do, Jeollanam-do, and Gyeongsang-nam-do in 2019 (Fig 3C and S3 Table).

Joinpoint analysis indicated that the rate per 1000 persons of national ER/LA opioid prescriptions increased annually by 64.1% (95% CI, 46.2%-84.1%; $P < 0.001$) from 2009 to 2014 and 5.9% (95% CI, 0.7. -11.3%; $P < 0.001$) from 2014 to 2019 (S3 Table). This pattern was observed in all administrative districts (Fig 3C and S3 Table). Among all administrative districts, Incheon showed the largest increase in the average APC, namely 36.6% (95% CI, 28.2%-45.6%, $P < 0.001$) (S3 Table).

### Comparison with US data

To identify whether the trends of prescribing opioids in South Korea is appropriate, we retrieved US data from studies by Scheiber et al. [11] and Kenan et al. [14], From a previous study, we abstracted the data of the rate per 100 persons regarding total and ER/LA opioid prescriptions. To directly compare the rate of prescription with our data, the rate was converted from per 100 persons to per 1000 persons (S4 Table). Notably, the rate of prescription per 1000 persons of ER/LA opioid prescriptions was higher in South Korea than in the US, that is, 53.5 and 76.2 in 2017 for the US and South Korea, respectively. The rate of prescription per 1000 persons of total opioids was lower in South Korea than in the US, that is, 588.5 and 499.7 in 2017 for the US and South Korea, respectively. To compare the rate of prescription of strong opioids, we systematically searched the literature. However, the data in 2009 by Kenan et al. [14] was the most recent data available. In their work, the data of opioid prescriptions that met our definition of strong opioids were retrieved and converted to the rate per 1000 persons. The rate of strong opioid prescriptions per 1000 was 206.7 in 2009 in the US (S5 Table). In South Korea, the highest rate of strong opioid prescriptions per 1000 persons was 15.2 in 2019.

### Discussion

This study demonstrated opioid prescribing trends nationally and for administrative districts across three key measures. The risk factors of opioid use disorder, overdose, and death are the prescription of strong opioids [15–17] and formulation of ER/LA opioids [18–21]. Based on these findings, we investigated the rate of prescriptions for total opioids, strong opioids, and ER/LA opioids. The annual rate of total, strong, and ER/LA opioid prescriptions per 1000 persons nationally increased over the 11-year period of investigation. The increase in

prescriptions of total opioids is largely attributed to an increase in the prescription of weak opioids. The prescriptions for strong opioids had the greatest percentage increase; however, it accounts for a small proportion of total opioid prescriptions.

Opioid outpatient prescriptions' growth was steep until 2013, both nationally and in most administrative districts, however, this growth declined between 2013 and 2019. Since propofol related mortality and misuse became an issue in South Korea, it was classified as a psychotropic agent and regulated from 2011. In 2012, the Ministry of Health and Welfare and the Ministry of Food and Drug Safety in South Korea announced a regulation to strengthen the management of all stages of manufacturing, distribution, and prescription to prevent the misuse of opioids and psychotropic drugs [22]. Due to Drug Utilization Review's (DUR) improvement, drugs in the same efficacy group and those with the same ingredients can be confirmed in the DUR, so that drugs are not prescribed excessively or duplicated. These strict government regulations seem to have caused the decrease in the opioid outpatient prescriptions' growth rate in 2013.

Our data also indicates that there is some cause for caution as it relates to opioid prescription. Disparities in the three key measures among administrative districts were apparent. The 90th percentile value of prescriptions was almost twice that of the 10th percentile for all three measures in 2019. Regarding the strong opioids, the regional difference was 2.5 times that of the 10th percentile in 2019. We visualized the intensity of the prescription rate on the map of Korea (Fig 2), therefore, areas of high intensity were easily recognized. Tertiary hospitals including cancer treatment centers located in densely populated metropolitan cities, especially Seoul in South Korea. However, the common administrative district with a high prescription rate per 1000 persons of total, strong, and ER/LA opioids was Jeollabuk-do. This may indicate that this area requires opioid education for both prescribers and pharmacists, and enhanced regulations.

By comparing our findings with data from the US, we found that strong opioids occupy a larger portion of the total opioids in the US than in South Korea. The rate of prescription per 1000 persons of total opioids and strong opioids were 588.5 in 2017 and 206.7 in 2009 in the USA; whereas, in South Korea, they were 499.7 in 2017 and 15.2 in 2019. Therefore, it can be safely postulated that our government regulations regarding opioid prescriptions has effectively prevented the excessive prescription of strong opioids. The rate of ER/LA prescriptions in South Korea was higher than that in the US. However, due to the level of our data, we were not able to calculate ER/LA prescriptions' MME/day. Thus, the higher ER/LA prescription rate in South Korea, than that in USA, was not sufficient to provoke the policymakers to create regulations on ER/LA opioid prescriptions in South Korea. Further studies are required to access the individual drug data level to calculate ER/LA opioids' MME/day.

Misuse and death from prescribed opioids have not yet been raised as an issue in Korea. However, Korea has experienced misuse of sedation drugs, such as propofol. Propofol was illegally prescribed and used for recreational purposes. Many deaths have occurred due to propofol overdose [23]. After social issues were raised, propofol was treated as an opioid and subsequently regulated by the government. If prescribed opioids were prevalent in Korea, the problem could be more serious than what was experienced with propofol. Many studies have demonstrated that an addiction to illicit opioids, such as heroin and fentanyl, may originate from an exposure to a high dose of prescribed opioids [24, 25]. Strict regulation related to opioid prescriptions are required at the government level before it becomes a public health problem.

This study has several limitations. We could not access the information of individual drugs because the information regarding which one of the three pharmaceutical companies manufactured the drug was not provided by the Health Insurance Review and Assessment Service.

Many strong opioids, such as oxycodone, were provided by a single pharmaceutical company in Korea. To evaluate the dosage of prescribed opioids, the MME should be calculated. Due to the limited access to data, we could not calculate the MME per prescription. In addition, this data contained no clinical information, including the reason opioids were prescribed, demographic information, or longitudinal data linking patients to clinical outcomes, such as opioid overdose morbidity and mortality.

## Conclusions

These data may inform government organizations about the necessity to create regulations and interventions for prescription opioids. The increase across all key measures of prescribed opioids and distinct geographic inequalities are significant findings. To monitor prescription opioids more effectively, further studies should be performed by accessing the data of individual opioid drugs. Therefore, annual reporting about trends of opioid prescriptions, including the MME per prescription, and opioid related morbidity and mortality should be published regularly.

## Supporting information

**S1 Table. Trends in rate (per 1000 population) of all opioids prescribed in South Korea, 2009–2019.**
(DOCX)

**S2 Table. Trends in rate (per 1000 population) of strong opioids prescribed in South Korea, 2009–2019.**
(DOCX)

**S3 Table. Trends in rate (per 1000 population) of ER/LA opioids prescribed in South Korea, 2009–2019.**
(DOCX)

**S4 Table. The rate of prescriptions per 1000 persons in the United States, 2006–2017.**
(DOCX)

**S5 Table. The rate of prescriptions of strong opioids per 1000 persons in the United States, 2006–2009.**
(DOCX)

## Author Contributions

**Conceptualization:** Noo Ree Cho, Dai Sik Ko, Jung Ju Choi.

**Data curation:** Noo Ree Cho, Dongchul Lee, Ji Ro Kim, Dai Sik Ko, Jung Ju Choi.

**Formal analysis:** Noo Ree Cho, Dongchul Lee, Dai Sik Ko, Jung Ju Choi.

**Funding acquisition:** Noo Ree Cho, Dai Sik Ko.

**Investigation:** Dai Sik Ko.

**Methodology:** Noo Ree Cho, Dongchul Lee, Ji Ro Kim, Dai Sik Ko, Jung Ju Choi.

**Resources:** Ji Ro Kim.

**Validation:** Noo Ree Cho, Dai Sik Ko, Jung Ju Choi.

**Visualization:** Dai Sik Ko, Jung Ju Choi.

**Writing – original draft:** Noo Ree Cho, Dai Sik Ko.

**Writing – review & editing:** Young Jin Chang, Dongchul Lee, Ji Ro Kim.

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
