## [Decision Letter · Decision Letter 0]

18 Mar 2021

PONE-D-21-01681

Trends in opioid prescribing practices in South Korea, 2009-2019: are we safe from an opioid epidemic?

PLOS ONE

Dear Dr. Ko,

Thank you for submitting your manuscript to PLOS ONE. After careful consideration, we feel that it has merit but does not fully meet PLOS ONE’s publication criteria as it currently stands. Therefore, we invite you to submit a revised version of the manuscript that addresses the points raised during the review process.

We look forward to receiving your revised manuscript.

Kind regards,

Vijayaprakash Suppiah, PhD

Academic Editor

PLOS ONE

Journal Requirements:

2. We note that Figure 2 in your submission contains map images which may be copyrighted. All PLOS content is published under the Creative Commons Attribution License (CC BY 4.0), which means that the manuscript, images, and Supporting Information files will be freely available online, and any third party is permitted to access, download, copy, distribute, and use these materials in any way, even commercially, with proper attribution. For these reasons, we cannot publish previously copyrighted maps or satellite images created using proprietary data, such as Google software (Google Maps, Street View, and Earth). For more information, see our copyright guidelines: http://journals.plos.org/plosone/s/licenses-and-copyright.

(1) You may seek permission from the original copyright holder of Figure 2 to publish the content specifically under the CC BY 4.0 license. 

3. In the Methods section, please provide further details on how opioid drugs were stratified for analysis.

Reviewers' comments:

Reviewer's Responses to Questions

**Comments to the Author**

1. Is the manuscript technically sound, and do the data support the conclusions?

Reviewer #1: Partly

Reviewer #2: Partly

2. Has the statistical analysis been performed appropriately and rigorously? 

Reviewer #1: I Don't Know

Reviewer #2: Yes

3. Have the authors made all data underlying the findings in their manuscript fully available?

Reviewer #1: No

Reviewer #2: Yes

4. Is the manuscript presented in an intelligible fashion and written in standard English?

Reviewer #1: Yes

Reviewer #2: Yes

5. Review Comments to the Author

Reviewer #1: Manuscript Number: PONE-D-21-01681

Title: Trends in opioid prescribing practices in South Korea, 2009-2019: are we safe from an

opioid epidemic?

Summary

I appreciate the opportunity to review this interesting report. This is a nationwide cross-sectional study in Korea that have shown opioids outpatient prescriptions from 2009 through 2019 to ensure that Korea are indeed free from an opioid epidemic.

Major Strengths

This is an interesting article that first describe an 11-year trend of opioids prescription in South Korea using nationwide data.

Major Weakness

1. Please more specify the objectives of the study in the introduction

2.Please describe more specific information of opioids. In addition, I wonder why the fentanyl patch was not included.

3.The authors indicate that over half of South Korean residents (53%) were prescribed opioids at least once in 2019. I think it is overrated.

4. In my opinion, there need some flow chart that represents inclusion and exclusion of the samples.

5. There was no information on the definition of chronic opioids use.

Reviewer #2: 63 “no issue” clarify statement. references?

71 “free from epidemic; how is this defined

91 state definition of “strong” here, explain how this relates to MME/day and limits of not using MME/day

124 “each year” - actually the mean value, clarify this. Is this meaningful?

126 “The number of total opioid prescriptions in 2019 was equal to 53% of the population of South Korea.”

128 In 2012 to 2013 there is a large change. This is the most important issue that needs to be addressed in the revision: why did this occur? Was there a change in data used? Was there a change in governmental regulation of opioids? Was there a change of opioid availability? Were new opioid products being introduced into the market? Were some drugs reclassified as opioids? Was there a change in how much could be dispensed by pharmacies, hence more prescriptions for the same amount of opioid? And so forth… The issue is that the paper rests on the tenuous data on the number of prescriptions and there is a significant change at this time. This may not correlate with increasing MME/day or with abuse, misuse, or overdose deaths. A thorough explanation of the reasons for this change must be explored to justify any conclusions drawn.

130 Was Sejong-si an outlier in terms of opioid prescription that could account for per capita changes?

255 greatest “percentage” increase

264 access must be addressed. Were pharmacies and specialist prescribers distributed equally in all districts? Did, for example, high prescription districts have major cancer treatment centers that others did not, etc.

265-269 simply incorrect. ER/LA opioids in the USA are problematic because the MME/day is much higher in these preparations. It is not known if this is the case in South Korea or if the ER/LA prescriptions written in South Korea correlate with higher MME/day prescribed. Without knowing this, no conclusion can be drawn about the need for regulation

276 “addiction to illicit opioids”

278-279 Possibly true but this conclusion cannot be drawn from the data presented. Revise “is needed” to “should be considered”

280-298 The main driving issue for regulatory change in the USA was overdose deaths - morbidity and mortality are not addressed in this data, and it is suggested that a national database of such data be established if it does not exist.

6. PLOS authors have the option to publish the peer review history of their article (what does this mean?). If published, this will include your full peer review and any attached files.

Reviewer #1: No

Reviewer #2: **Yes: **ADRIAN BARTOLI MD

---

## [Author Response · Author response to Decision Letter 0]

3 Apr 2021

Editor's Comments

Comment 1: Please ensure that your manuscript meets PLOS ONE's style requirements, including those for file naming. The PLOS ONE style templates can be found at

Response: Thank you for your observation and feedback. We have thoroughly reviewed your style requirements and have formatted the manuscript accordingly. If there are any further issues, please let us know.

Comment 2: We note that Figure 2 in your submission contains map images, which may be copyrighted. All PLOS content is published under the Creative Commons Attribution License (CC BY 4.0), which means that the manuscript, images, and Supporting Information files will be freely available online, and any third party is permitted to access, download, copy, distribute, and use these materials in any way, even commercially, with proper attribution. For these reasons, we cannot publish previously copyrighted maps or satellite images created using proprietary data, such as Google software (Google Maps, Street View, and Earth). For more information, see our copyright guidelines: http://journals.plos.org/plosone/s/licenses-and-copyright.

(1) You may seek permission from the original copyright holder of Figure 2 to publish the content specifically under the CC BY 4.0 license.

Response: Thank you for your advice. We used the R package “Kormap” to download a map of South Korea that shows all the administrative districts. The “Kormap” R package is a transformation of the shape file of the South Korea's map from the Statistical Geographic Information Service for utilization in R. Its license is Korea Open Government License Type 1. It allows the document to be copied, redistributed, remixed, and transformed for any purpose, even commercially, which is very similar to CC-BY 4.0. We received permissions from both the Statistical Geographic Information Service and Kormap’s creator to it use in our work under CC-BY 4.0 license. We have attached these permissions and updated the method.

Line 124: “We used the R package, “Kormap,” to download South Korea's map that shows all the administrative districts. The “Kormap” package is a transformation of the shape file of South Korea's map from the Statistical Geographic Information Service for utilization in R. The “Tmap” R package was used to visualize the administrative districts' disparity in opioid prescriptions in South Korea.”

Comment 3: In the Methods section, please provide further details on how opioid drugs were stratified for analysis.

Response: Thank you for your kind comment. We have updated the manuscript as advised.

Reviewers' Comments:

Reviewer #1: Manuscript Number: PONE-D-21-01681

Title: Trends in opioid prescribing practices in South Korea, 2009-2019: are we safe from an opioid epidemic?

Summary

I appreciate the opportunity to review this interesting report. This is a nationwide cross-sectional study in Korea that has shown opioids outpatient prescriptions from 2009 through 2019 to ensure that Korea is indeed free from an opioid epidemic.

Major Strengths

This is an interesting article that first describe an 11-year trend of opioids prescription in South Korea using nationwide data.

Major Weakness

Comment 4: Please more specify the objectives of the study in the introduction

Response: Thank you for your kind comment. We have specified the objectives in the introduction based on your feedback. Please see the inclusion below:

Line 68: "Accordingly, we aimed to examine the opioid outpatient prescription trends from 2009 through 2019 using the following strategies: (1) classification of opioids by potency and formula; (2) changes in the opioid outpatient prescriptions nationally each year; (3) geographical differences in the opioid outpatient prescriptions."

Comment 5: Please describe more specific information of opioids. In addition, I wonder why the fentanyl patch was not included.

Response: Thank you for your valuable feedback. We apologize for not explaining the meaning sufficiently. We have updated the manuscript by specifying the classification of opioids. The fentanyl patch has also been included in the revised manuscript. 

Line 93: “We defined strong opioids as those which are equivalent to or higher than morphine’s potency—the morphine milligram equivalents (MME) conversion factor is equal to or higher than 1 and is available in South Korea. The strong opioids were morphine, hydrocodone, fentanyl (including transdermal patches), hydromorphone, and oxycodone. The weak opioids were codeine, dihydrocodeine, tapentadol, and tramadol.”

Comment 6: The authors indicate that over half of South Korean residents (53%) were prescribed opioids at least once in 2019. I think it is overrated.

Response: Thank you for your comment. We agree with your statement that it is overrated and have, therefore, corrected it. Please see correction below:

Line 132: "The total opioid prescriptions in 2019 was equal to 53% of the South Korean population."

Comment 7: In my opinion, there need some flow chart that represents inclusion and exclusion of the samples.

Response: Thank you for your kind comment. We added the flowchart as Figure 1. Furthermore, based on your comment, we have improved its readability.

Comment 8: There was no information on the definition of chronic opioids use.

Response: Thank you for your feedback. We have updated the definition of chronic opioid use from reference 9 (Oh TK, Jeon Y-T, Choi JW. Trends in chronic opioid use and association with five-year survival in South Korea: a population-based cohort study. Br J Anaesth. 2019;123(5):655-63.)

Line 64: " A recent study reported that patients with chronic use (over 90 days of continuous supply) of weak and strong opioids increased between 2002 and 2015."

Reviewer #2:

Comment 9: Line 63 “no issue” clarify statement. References?

Response: Thank you for your advice. We have corrected it and added a reference.

Line 61: "The misuse of prescription opioids and related mortality have not been an issue in Korea [9]."

Comment 10: Line 71 “free from epidemic”; how is this defined.

Response: Thank you for your kind comment. We agreed that "free from epidemic" is subjective and hard to define. In addition, in the last paragraph of the introduction, it is unwise to not specify the objectives of our study. Therefore, we have corrected the last sentence of the introduction as follows:

Line 68: "Accordingly, we aimed to examine opioid outpatient prescription trends from 2009 through 2019 using the following strategies: (1) classification of opioids by potency and formula; (2) changes in the opioid outpatient prescriptions nationally each year; (3) geographical differences in the opioid outpatient prescriptions

Comment 11: Line 91 state definition of “strong” here, explain how this relates to MME/day and limits of not using MME/day

Response: Thank you for your advice. We had originally requested the Health Insurance Review and Assessment Service Database in South Korea for the number of prescriptions of individual drugs. The prescribing information of an individual drug is essential to calculate the MME/day. However, their policy prohibits them from sharing individual drug prescribing information. The policy indicates that prescribing data can be shared when more than four commercial companies provide the same drug with the same dose. In South Korea, many opioids are provided by individual companies, such as Pfizer and Janssen. Therefore, the only option to retrieve the opioid prescribing information is to categorize opioids into groups, so that there are over four companies.

To overcome this limitation, we classified opioids as strong and weak. We defined strong opioids as those which were equivalent to or higher than morphine’s potency, for which the MME conversion factor is equal to or higher than 1. We think we can identify the trends and changes in the opioid outpatients prescribing patterns using this strategy. Please consider our feedback in this regard.

Comment 12: Line 124 “each year” - actually the mean value, clarify this. Is this meaningful?

Response: We thank you for the feedback. We have adjusted the sentence based on your comments. We think this sentence can guide the readers to Table 1 and make it easier for them to understand our study.

Line 131: "From 2009 to 2019, an average of 23.2 million opioid prescriptions were issued from outpatient departments in South Korea."

Comment 13: Line 126 “The number of total opioid prescriptions in 2019 was equal to 53% of the population of South Korea.”

Response: Thank you for your kind correction. We agreed that it is overrated and have, therefore, corrected it accordingly.

Comment 14: Line 128 In 2012 to 2013 there is a large change. This is the most important issue that needs to be addressed in the revision: why did this occur? Was there a change in data used? Was there a change in governmental regulation of opioids? Was there a change of opioid availability? Were new opioid products being introduced into the market? Were some drugs reclassified as opioids? Was there a change in how much could be dispensed by pharmacies, hence more prescriptions for the same amount of opioid? And so forth… The issue is that the paper rests on the tenuous data on the number of prescriptions and there is a significant change at this time. This may not correlate with increasing MME/day or with abuse, misuse, or overdose deaths. A thorough explanation of the reasons for this change must be explored to justify any conclusions drawn.

Response: Thank you for your kind comment. We have updated the discussion, substantially.

Line 262: “Opioid outpatient prescriptions’ growth was steep until 2013, both nationally and in most administrative districts, however, this growth declined between 2013 and 2019. Since propofol related mortality and misuse became an issue in South Korea, it was classified as a psychotropic agent and regulated from 2011. In 2012, the Ministry of Health and Welfare and the Ministry of Food and Drug Safety in South Korea announced a regulation to strengthen the management of all stages of manufacturing, distribution, and prescription to prevent the misuse of opioids and psychotropic drugs [21]. Due to Drug Utilization Review’s (DUR) improvement, drugs in the same efficacy group and those with the same ingredients can be confirmed in the DUR, so that drugs are not prescribed excessively or duplicated. These strict government regulations seem to have caused the decrease in the opioid outpatient prescriptions’ growth rate in 2013.”

Comment 15: Line 130 Was Sejong-si an outlier in terms of opioid prescription that could account for per capita changes?

Response: We appreciate your feedback regarding this. The opioid outpatient prescription trends in Sejong-si are different from other administrative districts. It decreases by years (APC -7 from 2012 to 2019). However, the rate of total opioid prescription per 1000 persons in 2019 was 300.4, which is lower than other regions.

We had originally omitted the data from Sejong-si. However, the trends (decrease in the growth rate of total, strong, and ER/LA opioids from 2013) were not different nationally when the analysis was performed without the data from Sejong-si. Since this study's primary purpose is to help the government create regulations related to opioids, we decided not to omit the data from Sejong-si.

Comment 16: Line 255 greatest “percentage” increase

Response: Thank you for your kind comment. We have corrected the sentence.

Line 259: “The prescriptions for strong opioids had the greatest percentage increase; however, it accounts for a small proportion of total opioid prescriptions.”

Comment 17: Line 264 access must be addressed. Were pharmacies and specialist prescribers distributed equally in all districts? Did, for example, high prescription districts have major cancer treatment centers that others did not, etc.

Response: Thank you for this observation. We agreed with your comment and have addressed the effect of discrete distribution of tertiary hospitals related to our results.

Line 280: “Tertiary hospitals including cancer treatment centers located in densely populated metropolitan cities, especially Seoul in South Korea. However, the common administrative district with a high prescription rate per 1000 persons of total, strong, and ER/LA opioids was Jeollabuk-do. This may indicate that this area requires opioid education for both prescribers and pharmacists, and enhanced regulations.”

Comment 18: Line 265-269 simply incorrect. ER/LA opioids in the USA are problematic because the MME/day is much higher in these preparations. It is not known if this is the case in South Korea or if the ER/LA prescriptions written in South Korea correlate with higher MME/day prescribed. Without knowing this, no conclusion can be drawn about the need for regulation

Response: Thank you for your feedback. We agree with you and have significantly revised the section as follows:

Line 286: “By comparing our findings with data from the US, we found that strong opioids occupy a larger portion of the total opioids in the US than in South Korea. The rate of prescription per 1000 persons of total opioids and strong opioids were 588.5 in 2017 and 206.7 in 2009 in the USA. Whereas in South Korea, they were 499.7 in 2017 and 15.2 in 2019. Therefore, it can be safely postulated that our government regulations regarding opioid prescriptions has effectively prevented the excessive prescription of strong opioids. The rate of ER/LA prescriptions in South Korea was higher than that in the US. However, due to the level of our data, we were not able to calculate ER/LA prescriptions’ MME/day. Thus, the higher ER/LA prescription rate in South Korea, than that in USA, was not sufficient to provoke the policymakers to create regulations on ER/LA opioid prescriptions in South Korea. Further studies are required to access the individual drug data level to calculate ER/LA opioids’ MME/day.”

Comment 19: Line 276 “addiction to illicit opioids”

Response: Thank you for pointing this out. We have corrected per your suggestion.

Comment 20: Lines 278-279 Possibly true but this conclusion cannot be drawn from the data presented. Revise “is needed” to “should be considered”

Response: Thank you for your kind suggestion. We have corrected it accordingly.

Comment 21: Lines 280-298 The main driving issue for regulatory change in the USA was overdose deaths - morbidity and mortality are not addressed in this data, and it is suggested that a national database of such data be established if it does not exist.

Response: Thank you and we agreed with your comments. We have updated the last paragraph of the discussion section, as follows:

Line 314: "In addition, this data contained no clinical information, including the reason opioids were prescribed, demographic information, or longitudinal data linking patients to clinical outcomes, such as opioid overdose morbidity and mortality."

---

## [Decision Letter · Decision Letter 1]

19 Apr 2021

Trends in opioid prescribing practices in South Korea, 2009-2019: Are we safe from an opioid epidemic?

PONE-D-21-01681R1

Dear Dr. Ko,

We’re pleased to inform you that your manuscript has been judged scientifically suitable for publication and will be formally accepted for publication once it meets all outstanding technical requirements.

Kind regards,

Vijayaprakash Suppiah, PhD

Academic Editor

PLOS ONE

Reviewers' comments:

Reviewer's Responses to Questions

**Comments to the Author**

1. If the authors have adequately addressed your comments raised in a previous round of review and you feel that this manuscript is now acceptable for publication, you may indicate that here to bypass the “Comments to the Author” section, enter your conflict of interest statement in the “Confidential to Editor” section, and submit your "Accept" recommendation.

Reviewer #1: All comments have been addressed

2. Is the manuscript technically sound, and do the data support the conclusions?

Reviewer #1: Yes

3. Has the statistical analysis been performed appropriately and rigorously? 

Reviewer #1: Yes

4. Have the authors made all data underlying the findings in their manuscript fully available?

Reviewer #1: (No Response)

5. Is the manuscript presented in an intelligible fashion and written in standard English?

Reviewer #1: (No Response)

6. Review Comments to the Author

Reviewer #1: (No Response)

7. PLOS authors have the option to publish the peer review history of their article (what does this mean?). If published, this will include your full peer review and any attached files.

Reviewer #1: No

---

## [Editor Report · Acceptance letter]

3 May 2021

PONE-D-21-01681R1 

Trends in opioid prescribing practices in South Korea, 2009-2019: Are we safe from an opioid epidemic? 

Dear Dr. Ko:

I'm pleased to inform you that your manuscript has been deemed suitable for publication in PLOS ONE. Congratulations! Your manuscript is now with our production department. 

Kind regards, 

on behalf of

Dr. Vijayaprakash Suppiah 

Academic Editor

PLOS ONE